energy/materials science/nanotechnology

thermoelectric, nanoscale porosity, power factor, figure of merit

**Authors for correspondence:**
Md. Shahriar A. Hossain
e-mail: md.hossain@uq.edu.au
Xiaolin Wang
e-mail: xiaolin@uow.edu.au

This article has been edited by the Royal Society of Chemistry, including the commissioning, peer review process and editorial aspects up to the point of acceptance.

# Enhancement of thermoelectric properties of La-doped SrTiO$_3$ bulk by introducing nanoscale porosity

Al Jumlat Ahmed[1], Sheik Md. Kazi Nazrul Islam[1], Ridwone Hossain[1], Jeonghun Kim[2,3], Minjun Kim[3], Motasim Billah[3], Md. Shahriar A. Hossain[3,4], Yusuke Yamauchi[2,3,5,6,7] and Xiaolin Wang[1,8]

[1]Institute for Superconducting and Electronic Materials (ISEM), Australian Institute of Innovative Materials (AIIM), University of Wollongong, North Wollongong, New South Wales 2500, Australia
[2]Key Laboratory of Eco-chemical Engineering, College of Chemistry and Molecular Engineering, Qingdao University of Science and Technology (QUST), Qingdao 266042, People's Republic of China
[3]Australian Institute for Bioengineering and Nanotechnology (AIBN), The University of Queensland, Brisbane, Queensland 4072, Australia
[4]School of Mechanical and Mining Engineering, Faculty of Engineering, and [5]School of Chemical Engineering, University of Queensland, St Lucia, Queensland 4072, Australia
[6]International Center for Materials Nanoarchitectonics (WPI-MANA), National Institute for Materials Science (NIMS), 1-1 Namiki, Tsukuba, Ibaraki 305-0044, Japan
[7]Department of Plant and Environmental New Resources, Kyung Hee University, 1732 Deogyeong-daero, Giheung-gu, Yongin-si, Gyeonggi-do 446-701, South Korea
[8]Key Laboratory of Renewable Energy Technologies for Buildings, Ministry of Education, School of Materials Science and Engineering, Shandong Jianzhu University, Jinan, 250101, Shandong, PR China

(ID) AJA, 0000-0001-6713-9320; SMKNI, 0000-0001-7272-2713

Electron-doped SrTiO$_3$ is a well-known *n*-type thermoelectric material, although the figure of merit of SrTiO$_3$ is still inferior compared with *p*-type metal oxide-based thermoelectric materials due to its high lattice thermal conductivity. In this study, we have used a different amount of the non-ionic surfactant F127 during sample preparation to introduce nanoscale porosities into bulk samples of La-doped SrTiO$_3$. It has been observed that the porosities introduced into the bulk sample significantly improve the Seebeck coefficient and reduce the thermal conductivity by the charge carrier and phonon scattering respectively. Therefore, there is an overall enhancement in the power factor (PF) followed by

a dimensionless figure of merit ($zT$) over a wide scale of temperature. The sample 20 at% La-doped SrTiO$_3$ with 600 mg of F127 surfactant (SLTO 600F127) shows the maximum PF of 1.14 mW m$^{-1}$ K$^{-2}$ at 647 K which is 35% higher than the sample without porosity (SLTO 0F127), and the same sample (SLTO 600F127) shows the maximum value of $zT$ is 0.32 at 968 K with an average enhancement of 62% in $zT$ in comparison with the sample without porosity (SLTO 0F127).

# 1. Introduction

More than 60% of the total energy produced worldwide is being wasted as heat. This leftover heat can be used for producing necessary electrical energy by thermoelectric (TE) materials [1–4]. The full potential of TE materials can be used by using them with other energy conversion technologies [5]. Thermoelectric performance of a material is assessed by the dimensionless figure of merit, $zT = S^2\sigma T/\kappa$, where, $S$, $\sigma$, $T$ and $\kappa$ are the thermopower (Seebeck coefficient, µV K$^{-1}$), the electrical conductivity (S m$^{-1}$), the absolute temperature (K) and the thermal conductivity (W m$^{-1}$K$^{-1}$), respectively [6–9]. The term $S^2\sigma$ is called the power factor (PF) of the thermoelectric material as well. For power generation application, it is even more important for a thermoelectric material to have improved PF than to have high efficiency, since most ubiquitous heat sources are free [10]. The relationship of the Seebeck coefficient to carrier concentration for a doped semiconductor can be expressed as

$$S = \frac{8\pi^2 k_B^2}{3eh^2} \, m^* T \left(\frac{\pi}{3n}\right)^{2/3},$$

where $k_B$ stands for Boltzmann constant, $e$ for electron charge, $h$ refers to Planck's constant, $m^*$ accounts for the effective mass of the carrier, $T$ is the absolute temperature and $n$ the carrier concentration [8]. The electrical conductivity varies proportionally with carrier concentration and carrier mobility, $\sigma = ne\mu$, where $\mu$ is the carrier mobility. The overall thermal conductivity of a material is the product of the thermal diffusivity, the heat capacity, and the material density, $\kappa = \alpha C_p\rho$, where $\alpha$, $C_p$ and $\rho$ are the thermal diffusivity, the heat capacity at constant pressure and the material density, respectively [11]. Thermal conductivity $\kappa$ has two components: $\kappa_{el}$ is the thermal conductivity from the movement of the electrons and the holes, and $\kappa_{ph}$ is the contribution from the movement of phonons through the lattice, $\kappa = \kappa_{ph} + \kappa_{el}$ [7]. From Wiedemann–Franz Law, it is perceivable that there is an increase in $\kappa_{el}$ with increasing electrical conductivity, $\sigma$, and temperature, $T$. $\kappa_{el} = LT\sigma$, where $L$ denotes the Lorenz number. Normally, $L$ is treated as a universal factor with the value of $2.44 \times 10^{-8}$ W $\Omega$ K$^{-2}$ for a degenerate semiconductor [7]. However, there is a significant deviation in the Lorenz number of non-degenerate semiconductors, where $L$ converges to $1.5 \times 10^{-8}$ W $\Omega$ K$^{-2}$ [12]. Since the electronic thermal conductivity $\kappa_{el}$ is related to the electrical conductivity and high electrical conductivity is a prerequisite for a TE material, the lattice thermal conductivity $\kappa_{ph}$ has to be reduced to lower the overall thermal conductivity. The $\kappa_{ph}$ can be characterized by $\kappa_{ph} = \frac{1}{3}C_v V l$, where the heat capacity ($C_v$) and the phonon velocity ($V$) are constant, so the $\kappa_{ph}$ mainly relies on the phonon mean free path (MFP) ($l$) [3]. It has been reported that nanoscopic pores in silicon thin film can suppress the lattice thermal conductivity to the amorphous limit [13]. A modelling study on nanoporous SiGe suggests that the enhancement of the Seebeck coefficient by scattering only low-energy electrons and a decrease in the lattice thermal conductivity can take place because of nanoscale porosity in the material, but high sample density is essential to prevent deterioration in the electrical conductivity [14]. The effects of mesoporous structure on the TE properties of doped SrTiO$_3$ thin film were investigated. The mesoporous structure suppresses the thermal conductivity and improves the Seebeck coefficient because of phonon and carrier scattering. The incorporation of Brij-S10 surfactant into doped SrTiO$_3$ film increases the $zT$ [15,16].

Conventional thermoelectric materials, for example, Bi$_2$Te$_3$, PbTe and Cu$_2$Se exhibit high thermoelectric performance, but these materials have some limitations such as poor lifetime at high temperature in air, limited sources and high toxicity. On the other hand, metal oxide-based TE materials have a high lifetime at high temperature, are low-cost and non-toxic, and have minimal impact on the environment [17]. Some $p$-type metal oxide-based TE materials, such as NaCo$_2$O$_4$, layer-structured cobalt oxide and BiSeCuO, exhibit excellent TE properties. The highest value of the figure of merit reaches unity ($zT = 1.4$) for Bi$_{0.875}$Ba$_{0.125}$CuSeO [18]. As compared with the $p$-type oxides, $n$-type oxide materials have lower thermoelectric performance. To fabricate a TE module based on oxide materials, the $zT$ of $n$-type oxides should be improved to the level of $p$-type materials.

Strontium titanate (denoted as SrTiO$_3$) is a well-known $n$-type thermoelectric material with the cubic perovskite ABO$_3$ crystal structure having the lattice parameter of 0.3905 nm. The melting temperature of SrTiO$_3$ is 2080°C which gives it chemical as well as thermal stability at high temperature. The lattice thermal conductivity of SrTiO$_3$ is high, 12 W m$^{-1}$ K$^{-1}$ at room temperature [11]. SrTiO$_3$ with appropriate stoichiometry is an insulator having a bandgap of 3.2 eV. However, the electrical conductivity can be changed from insulating to metallic by substitutional doping of SrTiO$_3$ with La$^{3+}$ or Nb$^{5+}$. Several methods are available to tune material properties such as chemical doping, pressure, solid-state reaction and so on. However, chemical doping seems to be an effective approach to improve the material performance without physical damage of the material [9,19–22]. It has been reported that high $zT$ has been achieved for La-doped SrTiO$_3$ by creating a defective perovskite lattice containing A- and O-site vacancies with mixed-valence Ti$^{3+}$ and Ti$^{4+}$ in the B-sites [23,24]. The effects of spark plasma sintering (SPS) time on thermoelectric properties of lanthanum-doped SrTiO$_3$ have also been reported [25]. In another report, La-doped SrTiO$_3$ nanostructured bulk has been synthesized by SPS from chemically synthesized colloidal nanocrystals [26]. It has been reported that the addition of nanosized Ag metal particles in Sr$_{0.9}$La$_{0.1}$TiO$_3$ causes an increase in the carrier concentration and that the electrical connection is built into an Ag particle between the grains. They improve the electrical conductivity and reduce the thermal conductivity [27]. The thermoelectric properties of Sr$_{1-x}$La$_x$TiO$_3$ nanoparticle compacts which are fabricated by the hydrothermal process followed by cold pressure were investigated [28]. The morphology of the nanoparticle compacts had abundant interfaces, which effectively reduced phonon's MFP.

So far, the effect of nanoscale porosity on the TE properties of La-doped SrTiO$_3$ bulks has not been published. Here, it is reported for the first time that nanoscale porosity in a La-doped SrTiO$_3$ bulk sample has a significant impact on its thermoelectric properties. The nanoscale porosity suppresses the thermal conductivity and significantly enhances the Seebeck coefficient by the phonon and carrier scattering respectively. Therefore, there is an overall improvement in the PF and the $zT$ of La-doped SrTiO$_3$.

## 2. Results and discussion

The XRD patterns of La-doped SrTiO$_3$ calcined powders with different amounts of F127 surfactant are shown with respect to undoped SrTiO$_3$ in electronic supplementary material, figure S1. The XRD patterns match with Joint Committee on Powder Diffraction Standards (JCPDS) card number 00-001-1018, which confirms that the main phase is strontium titanium oxide (SrTiO$_3$). There are some impurity phases such as TiO$_2$ and SrTi$_{12}$O$_{19}$ [29] with peaks in the 2$\theta$ range of 25°–35° in the XRD patterns.

Electronic supplementary material, figure S2a,b shows the nitrogen absorption/desorption isotherms and pore size distributions of La-doped SrTiO$_3$ calcinated powder with different amounts of F127 surfactant, respectively. It is clear from the figures that absorption/desorption of nitrogen gas and pore volume increase with the amount of F127 surfactant, which indicates that the number of pores increases in the sample with the amount of surfactant. The average pore size is 8–9 nm in both the samples, which is in the range of mesoporosity (electronic supplementary material, table S1) since the pore diameter depends on the size of the micelle made by F127. The specific surface area and pore volume also increase with the amount of surfactant, which is also an indication that the pore number increases with the amount of surfactant (electronic supplementary material, table S1).

SEM images of the La-doped SrTiO$_3$ calcinated powders with different amounts of F127 surfactant also reveal that the number of pores increases with an increasing amount of F127 surfactant. The SLTO 0F127 sample (electronic supplementary material, figure S2c) has no porosity because it has zero amount of surfactant. There are a few mesoscale pores in the SLTO 200F127 powder (electronic supplementary material, figure S2d). The SLTO 600F127 powder has more mesopores compared with the other samples (electronic supplementary material, figure S2e). The SEM images also show that some pores have become agglomerated, which is because of the high calcination temperature [30].

Figure 1$b$ shows the powder XRD patterns of the La-doped SrTiO$_3$ bulks with different amounts of F127 in comparison with undoped SrTiO$_3$. There is no impurity phase that is detectable in the XRD patterns. The enlarged (200) and (211) diffraction peaks (figure 1$c$,$d$) are clearly shifted to a higher angle. This indicates that La$^{3+}$ has been successfully replaced on Sr$^{2+}$ sites in the crystal lattice of SrTiO$_3$ and it is because the La ion has a fixed 3+ valence and La ion has a smaller ionic radius of 1.36 Å than that of Sr$^{2+}$ (1.44 Å) [31]. In figure 1$c$,$d$, the peaks are K$\alpha_1$ and K$\alpha_2$ doublets rather than single peaks [32]. The extracted lattice parameters from the XRD patterns also show that the shrinkage

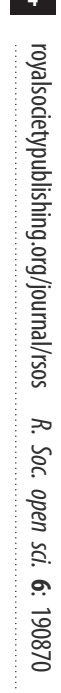

**Figure 1.** (*a*) Cubic perovskite crystal structure of SrTiO₃, (*b*) XRD patterns of La-doped SrTiO₃ bulk samples with different amounts of F127 surfactant in comparison with undoped SrTiO₃. (*c*) Enlarged (200) diffraction peak, (*d*) Enlarged (211) diffraction peak. The enlarged peaks are Kα₁ and Kα₂ doublets rather than single peaks [32].

in the lattice is caused by the La doping. The shrinkage in lattice is from 0.3901 nm for undoped SrTiO₃ to 0.389 for 20 at% La doping. The lattice parameter of samples is given in electronic supplementary material, table S2.

The density of the SLTO-F127 samples is listed in electronic supplementary material, table S3. The density of samples decreases slightly with increasing amounts of F127 surfactant in the sample. Since all the samples are sintered under the same sintering conditions, the reduction in density is an indication of the change in porosity inside the samples. The SEM cross-sectional images of the bulk samples (figure 2) also reveal that the number and size of the pores inside the samples change with the amount of F127 surfactant. The SLTO 0F127 sample (figure 2*a,d*) has no porosity, and the grains have become agglomerated. In figure 2*b,e*, the SLTO 200F127 sample has few nanoscale porosities in between grains. In the SLTO 600F127 sample, there are more mesopores in between grains compared with the sample SLTO 200F127, as shown in figure 2*c,f*. It can be also observed from figure 2 that the particle size increases with amount of surfactant in the sample. Increase in particle size could be the reason to keep the electrical conductivity unchanged, but, on the other hand, the porosities in between particles are responsible for phonon scattering which helps to reduce the phonon thermal conductivity.

The dependence on temperature of $\sigma$, $S$, PF and $\kappa$ for the La-doped SrTiO₃ samples with different amounts of F127 is shown in figure 3. For comparison, previously reported results [25,26,33,34] are shown in figure 3, as well. The undoped SrTiO₃ is found to be an insulator; however, its electrical conductivity has been improved with 20 at% La doping in each sample and this result is comparable with the previously reported results [23,24,26,35]. The electrical conductivity, $\sigma$ of all the samples increases initially with temperature up to 647 K (figure 3*a*), and then it starts to decrease with temperature afterward. There is no substantial change in $\sigma$ for the samples (figure 3*a*) with the different amount of F127 surfactant. The carrier concentration in all the samples is the same since the doping level is the same for all the samples.

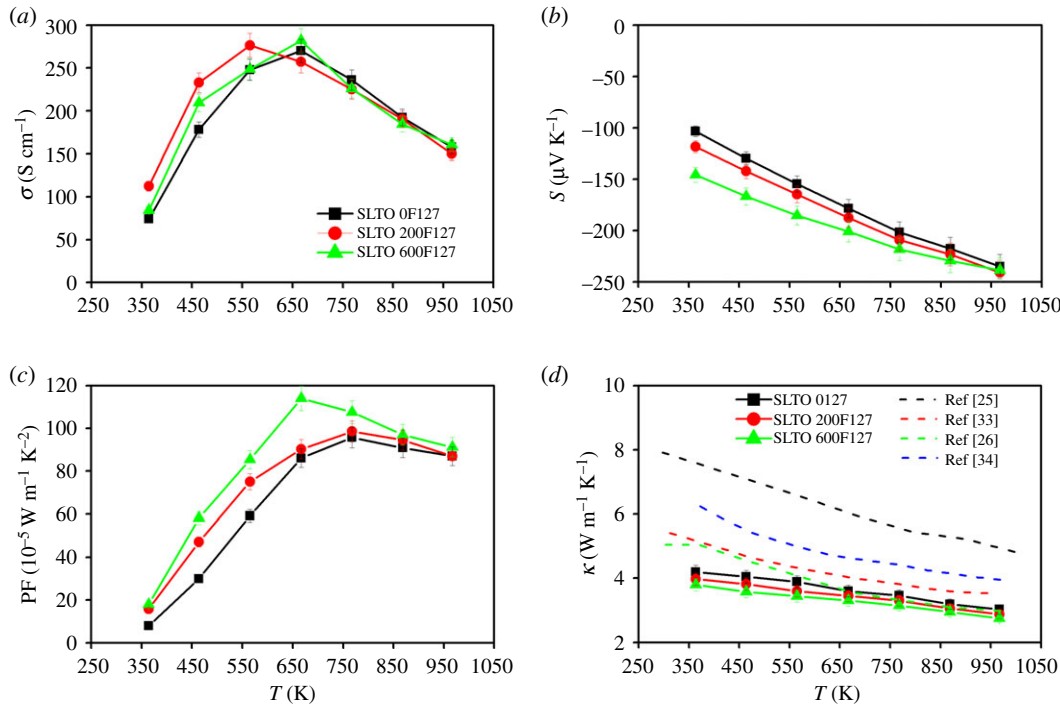

**Figure 2.** SEM cross-sectional images of bulk SrTiO$_3$ samples with different porosity: (*a*) SLTO 0F127, (*b*) SLTO 200F127, (*c*) SLTO 600F127, (*d*) high-resolution image of SLTO 0F127 of the selected region, (*e*) high-resolution image of SLTO 200F127 of the selected region, (*f*) high-resolution image of SLTO 600F127 of the selected region.

**Figure 3.** Temperature dependence of thermoelectric transport properties of the SLTO samples with different F127 surfactant: (*a*) the electrical conductivity ($\sigma$), (*b*) the Seebeck coefficient (*S*), (*c*) the PF and (*d*) the thermal conductivity ($\kappa$) in comparison with previously reported results [25,26,33,34].

The Seebeck coefficient, *S* of all the samples is negative, and it increases in magnitude with temperature (figure 3*b*). There is a significant improvement in the Seebeck coefficient with increasing amounts of F127 surfactant which could be due to the scattering of charge carriers by the pore. The SLTO 600F127 sample shows a high Seebeck coefficient compared with other samples over a wide

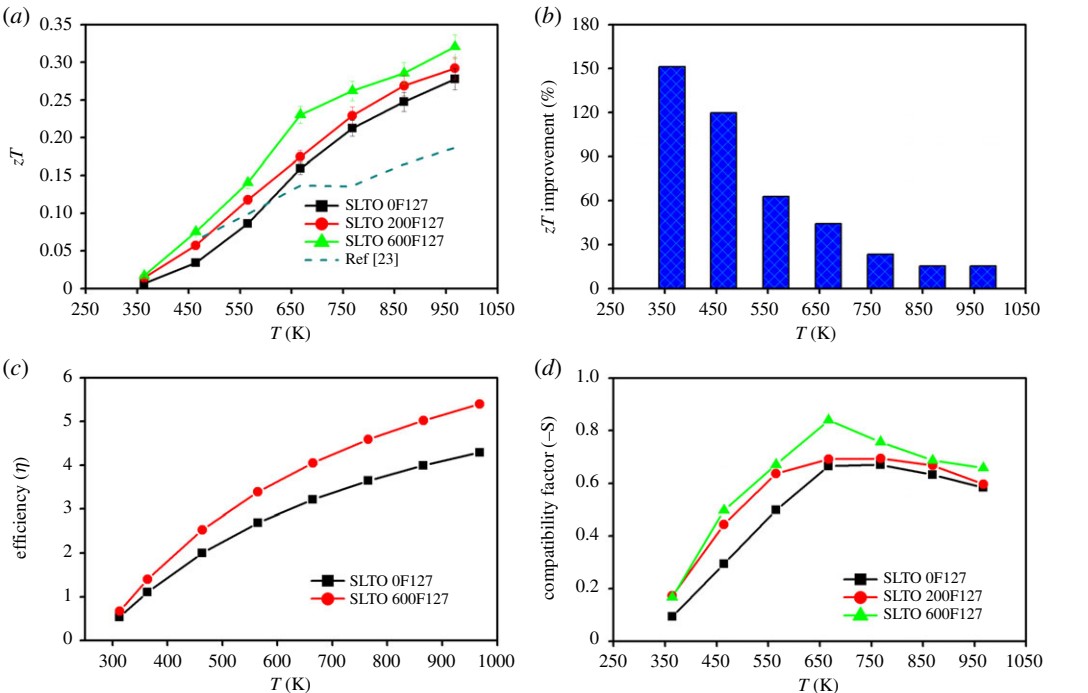

**Figure 4.** (*a*) The dimensionless figure of merit (*zT*) in comparison with published result of 20 at% La-doped SrTiO$_3$ [23], (*b*) the improvement in *zT* of the sample SLTO 600F127 compared with the sample SLTO 0F127, (*c*) the efficiency in percentage of samples and (*d*) the compatibility factor (−S) of the samples with different amount of surfactant F127.

scale of temperature. The maximum value of the Seebeck coefficient for this sample is 140 µV K$^{-1}$ at 325 K which is 52% higher than the sample without porosity (SLTO 0F127).

The PF for the samples is presented in figure 3*c*. Owing to the improvement in the Seebeck coefficient, there is a significant improvement in the PF also. The PF of the samples increases with the temperature up to 647 K, where it has its peak value. The SLTO 600F127 sample shows the highest value of the PF, 1.14 mW m$^{-1}$ K$^{-2}$ at 647 K, which is 35% higher than the PF of the sample without porosity (SLTO 0F127).

Figure 3*d* exhibits the change in the thermal conductivity, $\kappa$ of the samples with temperature. The thermal conductivity of the SLTO 0F127, SLTO 200F127 and SLTO 600F127 is found to be 3.03, 2.88 and 2.75 W m$^{-1}$ K$^{-1}$, respectively, at temperature 967 K. The gradual reduction in thermal conductivity has been observed with an increasing amount of surfactant. Moreover, the $\kappa$ of all the samples is found to be significantly lower than most of the published results [25,26,33,34]. The electronic thermal conductivity and the phonon thermal conductivity are presented in electronic supplementary material, figure S3a,b, respectively. The reduction in total thermal conductivity over the wide scale of temperature is due to the scattering of phonons by the nanoscale porosity. The electronic supplementary material, figure S3b is the evidence of a reduction in $\kappa_{Ph}$ due to the nanoscale porosity.

Because of the substantial improvement in the Seebeck coefficient and reduction in the thermal conductivity, there is an overall improvement in the *zT* of the porous samples (SLTO 200F127 and SLTO 600F127) compared with the non-porous sample SLTO 0F127 as shown in figure 4*a*. There is also a substantial improvement in the *zT* over a wide scale of temperature compared with the previously reported result of 20 at% La-doped SrTiO$_3$ [23]. The SLTO 600F127 sample shows the highest value of *zT* of 0.32 at 968 K.

The improvement in *zT* of the SLTO 600F127 in percentage compared with the sample SLTO 0F127 is shown in figure 4*b*. It is important to mention that the average improvement of *zT* in the SLTO 600F127 is found to be 62% compared with the SLTO 0F127. The efficiency of samples is calculated according to the literature [36]. The efficiency of samples SLTO 0F127 and SLTO 600F127 compared with the reported results is shown in figure 4*c* [23]. It is found that the efficiency for the SLTO 600F127 is greater than 5% at 968 K, which is around 26% higher than the SLTO 0F127 sample. The compatibility factor is important to cascade a thermoelectric material with another one to fabricate the segmented thermoelectric device. Two thermoelectric materials with a close compatibility factor are suitable for cascading. The compatibility factor of samples (SLTO 0F127, SLTO 200F127, SLTO 600F127) for the segmented thermoelectric generator is calculated based on the literature [37] as shown in figure 4*d*. It could help to find the suitable thermoelectric material for cascading with La-doped SrTiO$_3$.

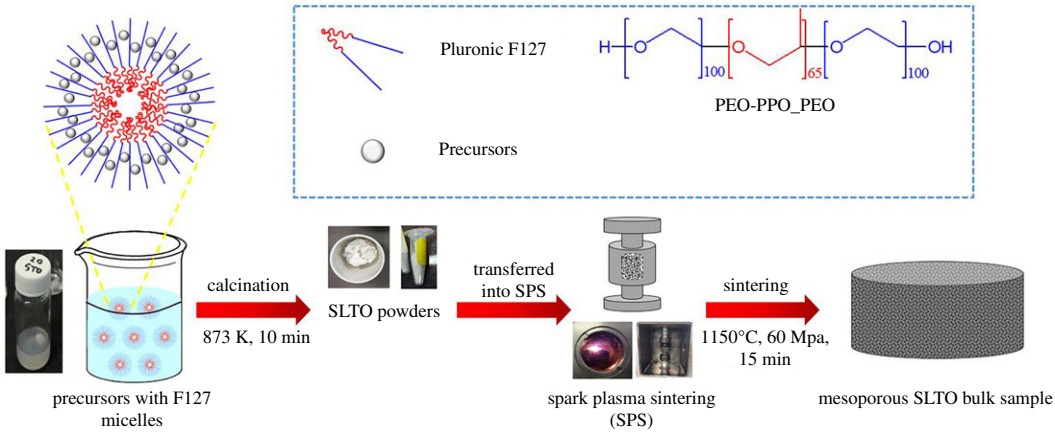

**Figure 5.** Schematic illustration of fabrication processes for the La-doped SrTiO₃ (SLTO) bulk sample with nanoscale porosity using F127 surfactant.

# 3. Conclusion

La-doped SrTiO₃ bulk samples with F127 surfactant in different amounts have been fabricated for the first time and investigated successfully. The experiments reveal that there is an impact of nanoscale porosity on the transport properties of La-doped SrTiO₃. It has been observed that the Seebeck coefficient increases, while the thermal conductivity is reduced substantially by introducing porosity into the bulk sample because of the carrier and phonon scattering by the nanoscale pores. Therefore, there is an overall enhancement in the PF and the $zT$. The sample, SLTO 600F127, exhibits the highest value of the PF, $1.14$ mW m$^{-1}$ K$^{-2}$ at 647 K, which is 35% higher than for the sample without porosity (SLTO 0F127). The same sample (SLTO 600F127) also exhibits the maximum value of the $zT$ is 0.32 at 968 K with an average enhancement of 62% in $zT$ in comparison with the sample without porosity (SLTO 0F127).

# 4. Experimental

## 4.1. Synthesis of La-doped SrTiO₃ (SLTO) powders with nanoscale porosity

First, strontium acetate (0.26 g) and lanthanum acetate hydrate (0.11 g for 20 at% La doping) were dissolved into acetic acid solution (3.0 ml) at 323 K with stirring. After the solution was cooled down to room temperature, titanium butoxide (0.61 g) was further added to it. The commercially available poly (ethylene oxide)-*b*-poly (propylene oxide)-*b*-poly (ethylene oxide) type triblock copolymer, Pluronic F127 (PEO₁₀₆PPO₇₀PEO₁₀₆), was used as the soft-template. Pluronic F127 in different amounts (200 and 600 mg) was dissolved in ethanol solution (3.0 g) in another beaker. The above two solutions were mixed together to prepare the precursor solutions. Then, the precursor solutions were transferred onto filter paper. Finally, the filter papers were calcined for 10 min at 873 K to remove the templates (figure 5).

## 4.2. Preparation of La-doped SrTiO₃ bulk

In the second stage of fabrication (figure 5), highly dense bulk samples of 20 mm diameter and around 1.5 mm thick (electronic supplementary material, table S1) were prepared from the La-doped SrTiO₃ calcinated powders by SPS (Thermal Technology SPS model 10⁻⁴) for 15 min at 1423 K with 60 MPa pressure in vacuum. Using a cutting machine (Struers Accutom-50), the bulk samples were shaped into rectangular bars and round discs for transport measurement. La-doped SrTiO₃ samples with different amounts of F127 surfactant were denoted by the amount of surfactant, such as SLTO $x$F127 ($x = 0$, 200 and 600).

## 4.3. Sample characterization

The room temperature powder XRD patterns were determined by the X-ray diffractometry (Cu K$\alpha$, GBC MMA, $\lambda = 1.5418$ Å). XRD patterns were measured with a step size of 0.02° and speed of 2° per min from 10° to 80°. The morphologies and nanostructures of the powder and bulk samples were studied using

field-emission scanning electron microscopy (FE-SEM, JEOL 7500). The surface area and porosity of powder samples were inspected by Brunauer–Emmett–Teller and Barrett–Joyner–Halenda analysis of nitrogen absorption–desorption data collected on a Tristar 3020 system (Micrometrics Instrument Corporation) after degassing at 150°C overnight. The $S$ and the $\sigma$ were measured from room temperature to 968 K under vacuum using Ozawa RZ2001i. The thermal diffusivity was measured under vacuum conditions using the instrument, LINSEIS LFA 1000, and the specific heat was measured under argon atmosphere by DSC-204F1 Phoenix. The weight and dimensions of a rectangular sample were used to determine the sample density. The results of the samples are confirmed by repeating all the measurements several times.

Data accessibility. Data are available from Dryad Digital Repository: https://doi.org/10.5061/dryad.4s6mn56 [38].

Authors' contributions. A.J.A., S.M.K.N.I., R.H. and M.B. carried out laboratory work, participated in data analysis and participated in the design of the study and drafted the manuscript. J.K., M.K. and M.S.A.H. critically revised the manuscript. Y.Y. and X.W. conceived of the study, designed the study, coordinated the study and helped draft the manuscript. All authors gave final approval for publication and agree to be held accountable for the work performed therein.

Competing interests. We declare we have no competing interests.

Funding. This work was partially supported by the Australian Research Council (ARC) through a Discovery Project DP 130102956 (X.W.), an ARC Professorial Future Fellowship project (FT 130100778, X.W.) and a Linkage Infrastructure Equipment and Facilities (LIEF) grant (LE 120100069, X.W.). This research was also supported by the Global Connections Fund (Bridging grant scheme) of the Australian Academy of Technology and Engineering (ATSE) in 2019 (M.S.A.H. and Y.Y.).

Acknowledgements. Higher degree research of Al Jumlat Ahmed and Sheik Md Kazi Nazrul Islam is supported by the Endeavour Leadership Program of Australian Government. We are thankful to Yaser Rehman, Hamzeh Qutaish and Alexander Morlando for assistance with sample characterization for SEM analysis and BET analysis. This work was also performed in part at the Queensland node of the Australian National Fabrication Facility, a company established under the National Collaborative Research Infrastructure Strategy to provide nano and microfabrication facilities for Australian Researchers.

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
