## [Reviewer comments · Royal Society Open Science]

Review History

RSOS-190870.R0 (Original submission)

Review form: Reviewer 1

Is the manuscript scientifically sound in its present form?

Yes

Are the interpretations and conclusions justified by the results?

No

Is the language acceptable?

No

Do you have any ethical concerns with this paper?

No

Have you any concerns about statistical analyses in this paper?

No

Recommendation?

Accept with minor revision (please list in comments)

Comments to the Author(s)

This is an interesting study into the effect of surfactant on the porosity, and ultimately thermoelectric properties, of known La-doped SrTiO₃ ceramics.

The comments below are suggestions provided with the aim towards improving the publication:

1. Page 2, line 60. Change the word "careers" to "carriers".
2. Page 4, line 39. Correct the equation - the fraction should be 1/3, not 1/4.
3. Page 4, line 41. Mention of " k_{ph} " which is not defined up to this point. Do the authors mean k_{ph} ?
4. In a few places, the equation for SrTiO₃ needs correcting - Page 6, line 1; Page 10, line 46; Page 10, line 58.
5. Page 5, line 59. Language of sentence "In another report, La-doped SrTiO₃ bulk with nanostructured has been synthesised by SPS from chemically synthesized colloidal nanocrystals" needs to be improved. Consider changing to "In another report, La-doped SrTiO₃ nanostructured bulk ceramics have been synthesised by SPS from chemically synthesis colloidal nanocrystals".
6. Page 7, line 38 and Figure 2. Shifts in reflections observed through XRD can result from sample positional changes as well as changes to the lattice. Authors should extract lattice parameters from their XRD patterns to make their analyses more quantitative.
7. Page 7, line 59. Surely referring to Fig. 3d, and not 6d?
8. Page 8, line 26. It is interesting to see that there is minimal change in the electronic conductivity despite there being a significant change in Seebeck coefficient and could be more related to the microstructure of the processed materials. In relation to the comment below on Figure 3, the addition of surfactant appears to affect both porosity and particle size. Increased grain growth would reduce resistive contributions from grain boundaries, however, comparable electronic conductivities are observed. As electronic conductivity measured from bulk ceramics can depend on many factors such as processing, grain boundary contributions, carrier concentration, the reviewer suggests that the authors estimate carrier concentration through thermogravimetric analysis under oxygen and compare these values against the measured conductivities.
9. Page 8, line 33. The authors state that the observed improvement in Seebeck coefficient could be due to scattering of charge carriers by the pores. The Seebeck coefficient is a property highly dependent on intrinsic electronic band structure, while the electronic conductivity can be affected much more by extrinsic factors such as ceramic microstructure, pores and grain boundaries, however, this is not manifested in the measured electronic conductivity which are all comparable. In relation to point 8, could the authors comment on this?
9. Page 8, line 54. The authors should comment on how their value of PF for non-porous SLTO 0F127 compares against literature values.
10. Page 9, line 3. Units for values of thermal conductivity need to be given.
11. Page 9, line 42. A brief description of the compatibility factor would be helpful to the reader.

Figure 3. While it is clear that porosity increases with increasing F127 surfactant, it also appears that particle size increases with F127 surfactant amount, which is supported by sharper XRD line shapes shown in Figure 2. Could the authors comment on this?

Figure 4. Error bars should be included on all plots.

Figure 5(a). Error bars should be plotted on zT values.

Review form: Reviewer 2

Is the manuscript scientifically sound in its present form?

Yes

Are the interpretations and conclusions justified by the results?

No

Is the language acceptable?

Yes

Do you have any ethical concerns with this paper?

No

Have you any concerns about statistical analyses in this paper?

No

Recommendation?

Major revision is needed (please make suggestions in comments)

Comments to the Author(s)

Thermoelectrics has attracted growing interests due to potential energy applications. Oxide compounds hold special interests to research community due to their unique physical and chemical features. In this paper, the author introduces nano-porosity into La-SrTiO₃, where certain enhancement of PF and ZT was observed.

1. One serious question is if you check your fig 3, the particle size of SLTO 600F127>SLTO 200F127>SLTO 600F127, how do you know the properties enhancement is not due to changing of particle size and grain boundary?
2. Can you present lattice thermal conductivity? this would be useful to understand the effects of porosity.
3. Why does porosity have more effect at low temperature than the high temperature?
4. Figure 5C, very confused green line for reference 23. please modify it.
5. Many errors and language issue through the paper such as (1) page 2 line 60, charge careers ;(2) page 8 line 40,140 mico V/K should be -140 micro V/K.

Decision letter (RSOS-190870.R0)

01-Jul-2019

Dear Mr Ahmed:

Title: Enhancement of Thermoelectric Properties of La-doped SrTiO₃ Bulk by Introducing Nanoscale Porosity

Manuscript ID: RSOS-190870

Thank you for submitting the above manuscript to Royal Society Open Science. On behalf of the Editors and the Royal Society of Chemistry, I am pleased to inform you that your manuscript will be accepted for publication in Royal Society Open Science subject to minor revision in accordance with the referee suggestions. Please find the reviewers' comments at the end of this email.

The reviewers and handling editors have recommended publication, but also suggest some minor revisions to your manuscript. Therefore, I invite you to respond to the comments and revise your manuscript.

Because the schedule for publication is very tight, it is a condition of publication that you submit the revised version of your manuscript before 10-Jul-2019. Please note that the revision deadline will expire at 00.00am on this date. If you do not think you will be able to meet this date please let me know immediately.

Best wishes,
Dr Laura Smith

Publishing Editor, Journals

On behalf of the Subject Editor Professor Anthony Stace and the Associate Editor Professor Kim Jelfs.

RSC Associate Editor:
 Comments to the Author:
 (There are no comments.)

RSC Subject Editor:
 Comments to the Author:
 (There are no comments.)

Reviewer comments to Author:
 Reviewer: 1

Comments to the Author(s)

This is an interesting study into the effect of surfactant on the porosity, and ultimately thermoelectric properties, of known La-doped SrTiO₃ ceramics. The comments below are suggestions provided with the aim towards improving the publication:

1. Page 2, line 60. Change the word "careers" to "carriers".
2. Page 4, line 39. Correct the equation - the fraction should be 1/3, not 1/4.
3. Page 4, line 41. Mention of " k_{1} " which is not defined up to this point. Do the authors mean k_{ph} ?
4. In a few places, the equation for SrTiO₃ needs correcting - Page 6, line 1; Page 10, line 46; Page 10, line 58.
5. Page 5, line 59. Language of sentence "In another report, La-doped SrTiO₃ bulk with nanostructured has been synthesised by SPS from chemically synthesized colloidal nanocrystals" needs to be improved. Consider changing to "In another report, La-doped SrTiO₃ nanostructured bulk ceramics have been synthesised by SPS from chemically synthesis colloidal nanocrystals".
6. Page 7, line 38 and Figure 2. Shifts in reflections observed through XRD can result from sample positional changes as well as changes to the lattice. Authors should extract lattice parameters from their XRD patterns to make their analyses more quantitative.
7. Page 7, line 59. Surely referring to Fig. 3d, and not 6d?
8. Page 8, line 26. It is interesting to see that there is minimal change in the electronic conductivity despite there being a significant change in Seebeck coefficient and could be more related to the microstructure of the processed materials. In relation to the comment below on Figure 3, the addition of surfactant appears to affect both porosity and particle size. Increased grain growth would reduce resistive contributions from grain boundaries, however, comparable electronic conductivities are observed. As electronic conductivity measured from bulk ceramics can depend on many factors such as processing, grain boundary contributions, carrier concentration, the

reviewer suggests that the authors estimate carrier concentration through thermogravimetric analysis under oxygen and compare these values against the measured conductivities.

9. Page 8, line 33. The authors state that the observed improvement in Seebeck coefficient could be due to scattering of charge carriers by the pores. The Seebeck coefficient is a property highly dependent on intrinsic electronic band structure, while the electronic conductivity can be affected much more by extrinsic factors such as ceramic microstructure, pores and grain boundaries, however, this is not manifested in the measured electronic conductivity which are all comparable.

In relation to point 8, could the authors comment on this?

9. Page 8, line 54. The authors should comment on how their value of PF for non-porous SLTO 0F127 compares against literature values.

10. Page 9, line 3. Units for values of thermal conductivity need to be given.

11. Page 9, line 42. A brief description of the compatibility factor would be helpful to the reader.

Figure 3. While it is clear that porosity increases with increasing F127 surfactant, it also appears that particle size increases with F127 surfactant amount, which is supported by sharper XRD line shapes shown in Figure 2. Could the authors comment on this?

Figure 4. Error bars should be included on all plots.

Figure 5(a). Error bars should be plotted on zT values.

Reviewer: 2

Comments to the Author(s)

Thermoelectrics has attracted growing interests due to potential energy applications. Oxide compounds hold special interests to research community due to their unique physical and chemical features. In this paper, the author introduces nano-porosity into La-SiTiO₃, where certain enhancement of PF and ZT was observed.

1. One serious question is if you check your fig 3, the particle size of SLTO 600F127>SLTO 200F127>SLTO 600F127, how do you know the properties enhancement is not due to changing of particle size and grain boundary?

2. Can you present lattice thermal conductivity? this would be useful to understand the effects of porosity.

3. Why does porosity have more effect at low temperature than the high temperature?

4. Figure 5C, very confused green line for reference 23. please modify it.

5. Many errors and language issue through the paper such as (1) page 2 line 60, charge careers ;(2) page 8 line 40,140 mico V/K should be -140 micro V/K.

Author's Response to Decision Letter for (RSOS-190870.R0)

See Appendix A.

RSOS-190870.R1 (Revision)

Review form: Reviewer 1

Is the manuscript scientifically sound in its present form?

Yes

Are the interpretations and conclusions justified by the results?

No

Is the language acceptable?

Yes

Do you have any ethical concerns with this paper?

No

Have you any concerns about statistical analyses in this paper?

No

Recommendation?

Accept with minor revision (please list in comments)

Comments to the Author(s)

This paper is a fine example that shows how introducing porosity in bulk thermoelectric materials can reduce thermal conductivity leading to enhancement of figure of merit.

1. The authors show that there is improvement in Seebeck coefficient whilst electronic conductivity remains unchanged between samples with no surfactant and samples with surfactant. The improvement in Seebeck coefficient is attributed to increased carrier scattering from the increased sample porosity, however, no explanation is given as to why the conductivity is unchanged. If there is increased carrier scattering as the authors suggest, then surely this would be reflected in the measured electronic conductivity also? As the electronic conductivity of bulk materials is influenced by many factors; carrier concentration, carrier mobility, particle size, grain boundary contributions, then the authors need to estimate the carrier concentration for the three samples to strengthen their conclusions. Carrier concentration can be estimated through TGA, or by measuring mass change before and after oxidation. The particle size, and therefore grain boundary contributions, change depending on the amount of surfactant as shown in Fig. 3, so the carrier concentration should be investigated to see if it changes or remains the same.
2. Page 17, Line 24. Consistency in units is required. All other units are written in the form $W m^{-1} K^{-1}$ and not W/mK .
3. Page 10, Line 60. Correct the word "carries" to "carriers".

Decision letter (RSOS-190870.R1)

05-Aug-2019

Dear Mr Ahmed:

Title: Enhancement of Thermoelectric Properties of La-doped SrTiO₃ Bulk by Introducing Nanoscale Porosity
Manuscript ID: RSOS-190870.R1

Thank you for submitting the above manuscript to Royal Society Open Science. On behalf of the Editors and the Royal Society of Chemistry, I am pleased to inform you that your manuscript will be accepted for publication in Royal Society Open Science subject to minor revision in accordance with the referee suggestions. Please find the reviewers' comments at the end of this email.

The reviewers and handling editors have recommended publication, but also suggest some minor revisions to your manuscript. Therefore, I invite you to respond to the comments and revise your manuscript.

Because the schedule for publication is very tight, it is a condition of publication that you submit the revised version of your manuscript before 14-Aug-2019. Please note that the revision deadline will expire at 00.00am on this date. If you do not think you will be able to meet this date please let me know immediately.

Best wishes,

Dr Ellis Wilde
Publishing Editor, Journals

On behalf of the Subject Editor Professor Anthony Stace and the Associate Editor Professor Kim Jelfs.

RSC Associate Editor
Comments to the Author:
(There are no comments.)

RSC Subject Editor
Comments to the Author:
(There are no comments.)

Reviewer comments to Author:
Reviewer: 1

Comments to the Author(s)

This paper is a fine example that shows how introducing porosity in bulk thermoelectric materials can reduce thermal conductivity leading to enhancement of figure of merit.

1. The authors show that there is improvement in Seebeck coefficient whilst electronic conductivity remains unchanged between samples with no surfactant and samples with surfactant. The improvement in Seebeck coefficient is attributed to increased carrier scattering from the increased sample porosity, however, no explanation is given as to why the conductivity is unchanged. If there is increased carrier scattering as the authors suggest, then surely this would be reflected in the measured electronic conductivity also? As the electronic conductivity of bulk materials is influenced by many factors; carrier concentration, carrier mobility, particle size,

grain boundary contributions, then the authors need to estimate the carrier concentration for the three samples to strengthen their conclusions. Carrier concentration can be estimated through TGA, or by measuring mass change before and after oxidation. The particle size, and therefore grain boundary contributions, change depending on the amount of surfactant as shown in Fig. 3, so the carrier concentration should be investigated to see if it changes or remains the same.

2. Page 17, Line 24. Consistency in units is required. All other units are written in the form $W m^{-1} K^{-1}$ and not W/mK .

3. Page 10, Line 60. Correct the word "carries" to "carriers".

Author's Response to Decision Letter for (RSOS-190870.R1)

See Appendices B & C.

RSOS-190870.R2 (Revision)

Review form: Reviewer 1

Is the manuscript scientifically sound in its present form?

Yes

Are the interpretations and conclusions justified by the results?

Yes

Is the language acceptable?

Yes

Do you have any ethical concerns with this paper?

No

Have you any concerns about statistical analyses in this paper?

No

Recommendation?

Accept as is

Comments to the Author(s)

The authors have addressed the comments of the previous review. The paper is an interesting example of how altering microstructure and porosity can affect bulk thermoelectric properties.

Decision letter (RSOS-190870.R2)

02-Sep-2019

Dear Mr Ahmed:

Title: Enhancement of Thermoelectric Properties of La-doped SrTiO₃ Bulk by Introducing Nanoscale Porosity
Manuscript ID: RSOS-190870.R2

It is a pleasure to accept your manuscript in its current form for publication in Royal Society Open Science. The chemistry content of Royal Society Open Science is published in collaboration with the Royal Society of Chemistry.

On behalf of the Subject Editor Professor Anthony Stace and the Associate Editor Professor Kim Jelfs.

RSC Associate Editor:
Comments to the Author:
(There are no comments.)

RSC Subject Editor:
Comments to the Author:
(There are no comments.)

Reviewer(s)' Comments to Author:
Reviewer: 1

Comments to the Author(s)
The authors have addressed the comments of the previous review. The paper is an interesting example of how altering microstructure and porosity can affect bulk thermoelectric properties.

Appendix A

6 July 2019

Dr Laura Smith
Publishing Editor, Journals
Royal Society Open Science - Chemistry Editorial Office
Royal Society of Chemistry
Thomas Graham House, Science Park, Milton Road, Cambridge, CB4 0WF, UK

Dr Laura Smith,

Thank you for your email regarding decision and reviewer comments on our manuscript, "Enhancement of Thermoelectric Properties of La-doped SrTiO₃ Bulk by Introducing Nanoscale Porosity" (Manuscript ID: RSOS-190870).

We are grateful to the respected reviewers for their valuable suggestions for further improvement of our manuscript. We have revised the manuscript in response to all the comments and suggestions raised by the reviewers and have highlighted the changes in color. We hope that the revised manuscript would be satisfactory to the reviewers.

In response to the comments and suggestions of reviewer: 1,

Comments to the Author(s):

This is an interesting study into the effect of surfactant on the porosity, and ultimately thermoelectric properties, of known La-doped SrTiO₃ ceramics.

Response: Thank you for your comment. It is nice to know that you have found our study interesting.

Suggestions:

1. Page 2, line 60. Change the word "careers" to "carriers".

Response: Correction has been done and highlighted the change in yellow colour

2. Page 4, line 39. Correct the equation - the fraction should be 1/3, not 1/4.

Response: The equation has been corrected and highlighted in yellow colour

3. Page 4, line 41. Mention of " k_{l} " which is not defined up to this point. Do the authors mean k_{ph} ?

Response: It is K_{ph} . Correction has been done and highlighted in yellow colour

4. In a few places, the equation for SrTiO₃ needs correcting - Page 6, line 1; Page 10, line 46; Page 10, line 58.

Response: Correction has been done and highlighted in yellow colour

5. Page 5, line 59. Language of sentence "In another report, La-doped SrTiO₃ bulk with nanostructured has been synthesised by SPS from chemically synthesized colloidal nanocrystals" needs to be improved. Consider changing to "In another report, La-doped SrTiO₃ nanostructured bulk ceramics have been synthesised by SPS from chemically synthesis colloidal nanocrystals".

Response: Language has been changed according to the suggestion of the reviewer.

6. Page 7, line 38 and Figure 2. Shifts in reflections observed through XRD can result from sample positional changes as well as changes to the lattice. Authors should extract lattice l

Response: Thank you for your important suggestion. The lattice parameter of undoped SrTiO₃ and La doped SrTiO₃ has been extracted from the XRD patterns. It has been added in the manuscript.

The extracted lattice parameters from the XRD patterns show that the shrinkage in lattice is happened by the La doping. The shrinkage in lattice from 0.3901 nm for undoped SrTiO₃ to 0.389 for 20 at% La doping.

Sample	Lattice Parameter (nm)
STO	0.3901
SLTO 0F127	0.3891
SLTO 200F127	0.3894
SLTO 600F127	0.3889

7. Page 7, line 59. Surely referring to Fig. 3d, and not 6d?

Response: Correction has been done and highlighted in yellow colour

8. Page 8, line 26. It is interesting to see that there is minimal change in the electronic conductivity despite there being a significant change in Seebeck coefficient and could be more related to the microstructure of the processed materials. In relation to the comment below on Figure 3, the addition of surfactant appears to affect both porosity and particle size. Increased grain growth would reduce resistive contributions from grain boundaries, however, comparable electronic conductivities are observed. As electronic conductivity measured from bulk ceramics can depend on many factors such as processing, grain boundary contributions, carrier concentration, the reviewer suggests that the authors estimate carrier concentration through thermogravimetric analysis under oxygen and compare these values against the measured conductivities.

Response: Thank you for the suggestion to estimate the carrier concentration through thermogravimetric (TGA) analysis. We tried to estimate the carrier concentration at room temperature in PPMS using the hall-effect measurement technique as this technique is followed in many literatures. However, we were unable to measure the carrier concentration of the samples because the electrical conductivity of samples at room temperature is lower than the minimum limit of the machine.

9. Page 8, line 33. The authors state that the observed improvement in Seebeck coefficient could be due to scattering of charge carriers by the pores. The Seebeck coefficient is a property highly dependent on intrinsic electronic band structure, while the electronic conductivity can be affected much more by extrinsic factors such as ceramic microstructure, pores and grain boundaries, however, this is not manifested in the measured electronic conductivity which are all comparable. In relation to point 8, could the authors comment on this?

Response: As we know that the electrical conductivity is directly proportional to carrier concentration and on the other hand the Seebeck coefficient is inversely proportional to carrier

concentration. It can be observed from the SEM images that the particle size and porosity increase with amount of surfactant in the sample. Our understanding is that increase in particle size is helpful to keep the electrical conductivity unchanged in the samples but the porosities in between of particles are responsible for carrier scattering which helps to improve the Seebeck coefficient.

9. Page 8, line 54. The authors should comment on how their value of PF for non-porous SLTO 0F127 compares against literature values.

Response: We would like to draw the attention that we haven't compared our value of PF with literature values. Although PF values of samples with different amount of surfactant have been compared and significant improvement in PF of the sample SLTO 600F127 has been observed compare to the sample without porosity, SLTO 0F127.

10. Page 9, line 3. Units for values of thermal conductivity need to be given.

Response: Unit of thermal conductivity has been added and highlighted by yellow colour.

11. Page 9, line 42. A brief description of the compatibility factor would be helpful to the reader.

Response: The following description has been added in the manuscript.

The compatibility factor is important to cascade a thermoelectric material with another one to fabricate segmented thermoelectric device. Two thermoelectric materials with close compatibility factor are suitable for cascading.

Figure 3. While it is clear that porosity increases with increasing F127 surfactant, it also appears that particle size increases with F127 surfactant amount, which is supported by sharper XRD line shapes shown in Figure 2. Could the authors comment on this?

Response: It is true from the fig 3 that the particle size increases with amount of surfactant in the sample. Our understanding is that increase in particle size is helpful to keep the electrical conductivity unchanged but on the other hand the porosities in between of particles are responsible for Phonon scattering which helps to reduce the phonon thermal conductivity.

Figure 4. Error bars should be included on all plots.

Response: Error bars have been included on all plots.

Figure 5(a). Error bars should be plotted on zT values.

Response: Error bars have been added on zT values.

The Response to comment and suggestions of Reviewer: 2

Comments to the Author(s)

Thermoelectrics has attracted growing interests due to potential energy applications. Oxide compounds hold special interests to research community due to their unique physical and chemical features. In this paper, the author introduces nano-porosity into La-SiTiO₃, where certain enhancement of PF and ZT was observed.

Response: Thank you for the comment.

Suggestions

1. One serious question is if you check your fig 3, the particle size of SLTO 600F127 > SLTO 200F127 > SLTO 600F127, how do you know the properties enhancement is not due to changing of particle size and grain boundary?

Response: Thank you for your interesting question. It is true from the fig 3 that the particle size increases with amount of surfactant in the sample. Our understanding is that increase in particle size and grain boundary could be helpful to keep the electrical conductivity unchanged and on the other hand porosities in between of particles are responsible for Phonon scattering which helps to reduce the phonon thermal conductivity.

2. Can you present lattice thermal conductivity? this would be useful to understand the effects of porosity.

Response: The lattice/phonon thermal conductivity shows that the nano scale porosities help to reduce the lattice/phonon thermal conductivity by phonon scattering. The lattice thermal conductivity has been added in the supplementary document.

Fig: Phonon thermal conductivity

Fig: Electrical thermal conductivity

3. Why does porosity have more effect at low temperature than the high temperature?

Response: Thank you for your observation. We can see that the improvement in Seebeck coefficient at low temperature is higher than the value of Seebeck coefficient at high temperature. But we are unable to explain it clearly. It could be related to carrier mobility and carrier concentration with temperature.

4. Figure 5C, very confused green line for reference 23. please modify it.

Response: The green line for ref 23 has been removed from the fig 5c.

5. Many errors and language issue through the paper such as (1) page 2 line 60, charge careers ;(2) page 8 line 40,140 mico V/K should be -140 micro V/K.

Response: Errors have been corrected and highlighted in yellow colour.

We hope the revised manuscript would be satisfactory to the reviewers.

Sincerely yours,

Professor XIAOLIN WANG, PhD, FAIP

Node Leader and Theme Leader,

ARC Centre of Excellence in Future Low-Energy Electronics Technologies

ARC Professorial Future Fellow

Director

Institute for Superconducting and Electronic Materials

Australian Institute of Innovative Materials

The University of Wollongong, AIIM Facility (Building 231), Innovation Campus. Squires
Way, North Wollongong, NSW, 2500.

Email: xiaolin@uow.edu.au

T: +61 2 4221 4558; Fax: +61 2 42215731

Appendix B

7 August 2019

Dr Ellis Wilde
Publishing Editor, Journals

Dr Ellis Wilde,

Thank you for your email regarding decision and reviewer comments on our manuscript, “Enhancement of Thermoelectric Properties of La-doped SrTiO₃ Bulk by Introducing Nanoscale Porosity” (Manuscript ID: RSOS-190870.R1).

We are grateful to the respected reviewer for his valuable suggestions for further improvement of our manuscript. We have revised the manuscript in response to all the comments and suggestions raised by the reviewer and have highlighted the changes in color. We hope that the revised manuscript would be satisfactory to the reviewer.

In response to the comments and suggestions of reviewer: 1,

Comments to the Author(s)

This paper is a fine example that shows how introducing porosity in bulk thermoelectric materials can reduce thermal conductivity leading to enhancement of figure of merit.

Response: Thank you for the comment.

1. The authors show that there is improvement in Seebeck coefficient whilst electronic conductivity remains unchanged between samples with no surfactant and samples with surfactant. The improvement in Seebeck coefficient is attributed to increased carrier scattering from the increased sample porosity, however, no explanation is given as to why the conductivity is unchanged. If there is increased carrier scattering as the authors suggest, then surely this would be reflected in the measured electronic conductivity also? As the electronic conductivity of bulk materials is influenced by many factors; carrier concentration, carrier mobility, particle size, grain boundary contributions, then the authors need to estimate the carrier concentration for the three samples to strengthen their conclusions. Carrier concentration can be estimated through TGA, or by measuring mass change before and after oxidation. The particle size, and therefore grain boundary contributions, change depending on the amount of surfactant as shown in Fig. 3, so the carrier concentration should be investigated to see if it changes or remains the same.

Response:

The change in mass before and after oxidation of the samples has been measured to estimate the carrier concentration. The samples are kept in a furnace at 1000°C for 2 hours in air for oxidation. The samples become partially white in colour after oxidation as shown in the fig b. The change in mass after oxidation of the samples is within a fraction of milligram. And there is not much difference in change of mass for the samples as listed in the table.

Fig. (a) The colour of samples is black before oxidation. (b) The samples are become partially white after oxidation at 1000°C for 2 hr.

Table: Change in mass of samples after oxidation

Sample	Mass before oxidation (mg)	Mass after oxidation (mg)	Change in Mass (mg)
SLTO 0F127	135.1	135.8	0.7
SLTO 200F127	123.1	123.9	0.8
SLTO 600F127	077.0	077.5	0.5

2. Page 17, Line 24. Consistency in units is required. All other units are written in the form $W m^{-1} K^{-1}$ and not W/mK .

Response: Correction is made. It is highlighted in green colour.

3. Page 10, Line 60. Correct the word "carries" to "carriers".

Response: Correction is made and highlighted in green colour.

We hope the revised manuscript would be satisfactory to the reviewers.

Sincerely yours,

Professor XIAOLIN WANG, PhD, FAIP

Node Leader and Theme Leader,

ARC Centre of Excellence in Future Low-Energy Electronics Technologies

ARC Professorial Future Fellow

Director

Institute for Superconducting and Electronic Materials

Australian Institute of Innovative Materials

The University of Wollongong, AIIM Facility (Building 231), Innovation Campus. Squires
Way, North Wollongong, NSW, 2500.

Email: xiaolin@uow.edu.au

T: +61 2 4221 4558; Fax: +61 2 42215731

Appendix C

Enhancement of Thermoelectric Properties of La-doped SrTiO₃ Bulk by Introducing Nanoscale Porosity

Al Jumlat Ahmed ^a, Sheik Md. Kazi Nazrul Islam ^a, Ridwone Hossain ^a, Jeonghun Kim ^{b,c}, Minjun Kim ^c, Motasim Billah ^c, Md. Shahriar A. Hossain ^{*c,d}, Yusuke Yamauchi ^{b,c,e,f,g} and Xiaolin Wang ^{*a}

* Corresponding authors

^a Institute for Superconducting and Electronic Materials (ISEM), Australian Institute of Innovative Materials (AIIM), University of Wollongong, North Wollongong, NSW 2500, Australia, E-mail: xiaolin@uow.edu.au

^b Key Laboratory of Eco-chemical Engineering, College of Chemistry and Molecular Engineering, Qingdao University of Science and Technology (QUST), Qingdao 266042, China

^c Australian Institute for Bioengineering and Nanotechnology (AIBN), The University of Queensland, Brisbane, QLD 4072, Australia, E-mail: md.hossain@uq.edu.au

^d School of Mechanical and Mining Engineering, Faculty of Engineering, Architecture and Information Technology (EAIT), University of Queensland, St Lucia QLD 4072, Australia

^e International Center for Materials Nanoarchitectonics (MANA), National Institute for Materials Science (NIMS), 1-1 Namiki, Tsukuba, Ibaraki 305-0044, Japan

^f School of Chemical Engineering, Architecture and Information Technology (EAIT), University of Queensland, St Lucia QLD 4072, Australia

^g Department of Plant & Environmental New Resources, Kyung Hee University, 1732 Deogyong-daero, Giheung-gu, Yongin-si, Gyeonggi-do 446-701, South Korea

Abstract

Electron doped-SrTiO₃ is a well-known *n*-type thermoelectric material, although the figure of merit of SrTiO₃ is still inferior compared to *p*-type metal oxide-based thermoelectric materials due to its high lattice thermal conductivity. In this study, we have used different amount of the non-ionic surfactant F127 during sample preparation to introduce nanoscale porosities into bulk samples of La-doped SrTiO₃. Thermopower is observed to be improved by the charge **carriers** due to introduction of porosity into the bulk sample which resulted in lowering thermal

conductivity due to enhanced phonon scattering. It has been observed that the porosities introduced into the bulk sample significantly improve the Seebeck coefficient and reduce the thermal conductivity by the charge carrier and phonon scattering respectively. Therefore, there is an overall enhancement in the power factor (PF) followed by dimensionless figure of merit (zT) over a wide scale of temperature. The sample 20 at% La-doped SrTiO₃ with 600 mg of F127 surfactant (SLTO 600F127) shows the maximum power factor of 1.14 mWm⁻¹K⁻² at 647 K which is 35% higher than the sample without porosity (SLTO 0F127), and the same sample (SLTO 600F127) shows the maximum value of zT is 0.32 at 968 K with an average enhancement of 62% in zT in comparison to the sample without porosity (SLTO 0F127).

Introduction

More than 60 percent of total energy produced worldwide is being wasted as heat. This leftover heat can be utilized for producing necessary electrical energy by thermoelectric (TE) materials¹⁻⁴. The full potential of TE materials can be utilized by using them with other energy conversion technologies⁵. Thermoelectric performance of a material is assessed by the dimensionless figure of merit, $zT = \frac{S^2\sigma T}{\kappa}$, where, S , σ , T , and κ are the thermopower (Seebeck coefficient, $\mu\text{V}/\text{K}$), the electrical conductivity (S/m), the absolute temperature (K), and the thermal conductivity (W/mK), respectively⁶⁻⁹. The term $S^2\sigma$ is called the PF of the Thermoelectric material as well. For power generation application, it is even more important for a Thermoelectric material to have improved power factor than to have a high efficiency, since most ubiquitous heat sources are free¹⁰. The relationship of the Seebeck coefficient to carrier concentration for doped semiconductor can be expressed as $S = \frac{8\pi^2 k_B^2}{3eh^2} m^* T \left(\frac{\pi}{3n}\right)^{2/3}$, where, k_B stands for Boltzmann constant, e for electron charge, h refers to Planck's constant, m^* accounts to effective mass of the carrier, T is the absolute temperature, and n the carrier concentration⁸. The electrical conductivity varies proportionally with carrier concentration and

carrier mobility, $\sigma = n e \mu$, where μ is the carrier mobility. The over-all thermal conductivity of a material is the product of the thermal diffusivity, the heat capacity, and the material density, $\kappa = \alpha C_p \rho$, Where, α , C_p , and ρ are the thermal diffusivity, the heat capacity at constant pressure, and the material density, respectively¹¹. Thermal conductivity κ has two components: κ_{el} is the thermal conductivity from movement of the electrons and the holes, and κ_{ph} is the contribution from the movement of phonons through the lattice, $\kappa = \kappa_{ph} + \kappa_{el}$ ⁷. From Wiedemann-Franz law, it is perceivable that an increase in κ_{el} with increasing electrical conductivity, σ , and temperature, T . $\kappa_{el} = LT\sigma$, where L denotes the Lorenz number. Normally, L is treated as a universal factor with the value of $2.44 \times 10^{-8} \text{ W}\Omega/\text{K}^2$ for a degenerate semiconductor⁷. However, there is a significant deviation in the Lorenz number of non-degenerate semiconductors, where L converges to $1.5 \times 10^{-8} \text{ W}\Omega/\text{K}^2$ ¹². Since the electronic thermal conductivity κ_{el} is related to the electrical conductivity and high electrical conductivity is a prerequisite for a TE material, the lattice thermal conductivity κ_{ph} has to be reduced to lower the overall thermal conductivity. The κ_{ph} can be characterized by, $\kappa_{ph} = \frac{1}{3} C_v V l$, where the heat capacity (C_v) and the phonon velocity (V) are constant, so the κ_{ph} mainly relies on the phonon mean free path (MFP) (l)³. It has been reported that nanoscopic pores in silicon thin film can suppress the lattice thermal conductivity to the amorphous limit¹³. A modelling study on nanoporous SiGe suggests that enhancement of the Seebeck coefficient by scattering only low-energy electrons and decrease in the lattice thermal conductivity can take place because of nanoscale porosity in the material, but high sample density is essential to prevent deterioration in the electrical conductivity¹⁴. The effects of mesoporous structure on the TE properties of doped SrTiO₃ thin film were investigated. The mesoporous structure suppresses the thermal conductivity and improves the Seebeck coefficient because of phonon and carrier scattering. The incorporation of Brij-S10 surfactant into doped SrTiO₃ film increases the zT ¹⁵,

Conventional Thermoelectric materials for example Bi_2Te_3 , PbTe and Cu_2Se exhibit high thermoelectric performance, but these materials have some limitations such as poor lifetime at high temperature in air, limited sources, and high toxicity. On the other hand, metal oxide based TE materials have high lifetime at high temperature, low-cost, non-toxic and have minimal impact on environment¹⁷. Some *p*-type metal oxide-based TE materials such as NaCo_2O_4 , layer-structured cobalt oxide, and BiSeCuO exhibit excellent TE properties. The highest value of figure of merit reaches unity ($zT = 1.4$) for $\text{Bi}_{0.875}\text{Ba}_{0.125}\text{CuSeO}$ ¹⁸. As compared to the *p*-type oxides, *n*-type oxide materials have lower thermoelectric performance. To fabricate a TE module based on oxide materials, the zT of *n*-type oxides should be improved to the level of *p*-type materials. Strontium titanate (denoted as SrTiO_3) is a well-known *n*-type Thermoelectric material with the cubic perovskite ABO_3 crystal structure having the lattice parameter of 0.3905 nm. The melting temperature of SrTiO_3 is 2080°C which gives it chemical as well as thermal stability at high temperature. The lattice thermal conductivity of SrTiO_3 is high, 12 W/mK at room temperature¹¹. SrTiO_3 with appropriate stoichiometry is an insulator having a band gap of 3.2 eV. However, the electrical conductivity can be changed from insulating to metallic by substitutional doping of SrTiO_3 with La^{3+} or Nb^{5+} . Several methods are available to tune material properties such as chemical doping, pressure, solid state reaction and so on. However, chemical doping seems to be an effective approach to improve the material performance without physical damage of the material^{9, 19-22}. It has been reported that high zT has been achieved for La-doped SrTiO_3 by creating a defective perovskite lattice containing A- and O- site vacancies with mixed valence Ti^{3+} and Ti^{4+} on the B-sites^{23, 24}. The effects of spark plasma sintering (SPS) time on Thermoelectric properties of Lanthanum doped SrTiO_3 have also been reported²⁵. In another report, La-doped SrTiO_3 nanostructured bulk has been synthesised by SPS from chemically synthesized colloidal nanocrystals²⁶. It has been reported that the addition of nanosized Ag metal particles in $\text{Sr}_{0.9}\text{La}_{0.1}\text{TiO}_3$ causes an increase in the

carrier concentration and that the electrical connection is built into Ag particle between the grains. They improve the electrical conductivity and reduce the thermal conductivity²⁷. The thermoelectric properties of $\text{Sr}_{1-x}\text{La}_x\text{TiO}_3$ nanoparticle compacts which is fabricated by the hydrothermal process followed by cold pressure were investigated²⁸. The morphology of the nanoparticle compacts had abundant interfaces, which effectively reduced phonon's mean free path.

So far, the effect of nanoscale porosity on the TE properties of La-doped SrTiO_3 bulks has not been published. Here, it is reported for the first time that nanoscale porosity in a La-doped SrTiO_3 bulk sample has a significant impact on its thermoelectric properties. The nanoscale porosity suppresses the thermal conductivity and significantly enhances the Seebeck coefficient by the phonon and carrier scattering respectively. Therefore, there is an overall improvement in the PF and the zT of La-doped SrTiO_3 .

Results and discussion

The XRD patterns of La-doped SrTiO_3 calcined powders with different amounts of F127 surfactant are shown with respect to undoped SrTiO_3 in Fig. S1. The XRD patterns match with Joint Committee on Powder Diffraction Standards (JCPDS) card number 00-001-1018, which confirms that the main phase is strontium titanium oxide (SrTiO_3). There are some impurity phases such as TiO_2 and $\text{SrTi}_{12}\text{O}_{19}$ ²⁹ with peaks in the 2θ range of $25^\circ - 35^\circ$ in the XRD patterns.

Fig. S2 a and b shows the nitrogen absorption/desorption isotherms and pore size distributions of La-doped SrTiO_3 calcinated powder with different amounts of F127 surfactant respectively. It is clear from the figures that absorption/desorption of nitrogen gas and pore volume increase with the amount of F127 surfactant, which indicates that the number of pores increases in the sample with the amount of surfactant. The average pore size is 8 to 9 nm in both the samples, which is in the range of mesoporosity (Table S1), since the diameter of a pore depends on the size of the micelle formed by the surfactant. The specific surface area and

pore volume also increase with the amount of surfactant, which is also an indication that the pore number increases with the amount of surfactant (Table S1).

SEM images of the La-doped SrTiO₃ calcinated powders with different amounts of F127 surfactant also reveal that the number of pores increases with an increasing amount of F127 Surfactant. The SLTO 0F127 sample (Fig. S2 c) has no porosity because it has zero amount of surfactant. There are a few mesoscale pores in the SLTO 200F127 powder (Fig. S2 d). The SLTO 600F127 powder has more mesopores compared to the other samples (Fig. S2 e). The SEM images also show that some pores have become agglomerated, which is because of the high calcination temperature³⁰.

Fig. 2b shows the powder XRD patterns of the La-doped SrTiO₃ bulks with different amounts of F127 in comparison with undoped SrTiO₃. There is no impurity phase that is detectable in the XRD patterns. The enlarged (200) and (211) diffraction peaks (Fig. 2c and 2d) are clearly shifted to a higher angle. This indicates that La³⁺ has been successfully replaced on Sr²⁺ sites in the crystal lattice of SrTiO₃ and it is because of the La ion has a fixed 3+ valence and La ion has smaller ionic radius of 1.36 Å than that of Sr²⁺ (1.44 Å)³¹. In Fig. 2c and 2d, the peaks are K α ₁ and K α ₂ doublets rather than single peaks³². The extracted lattice parameters from the XRD patterns also show that the shrinkage in lattice is happened by the La doping. The shrinkage in lattice from 0.3901 nm for undoped SrTiO₃ to 0.389 for 20 at% La doping. The lattice parameter of samples is given in Table S2.

The density of the SLTO-F127 samples is listed in Table S3. The density of samples decreases slightly with increasing amounts of F127 surfactant in the sample. Since all the samples are sintered under the same sintering conditions, the reduction in density is an indication of the change in porosity inside the samples. The SEM cross-sectional images of the bulk samples (Fig. 3) also reveal that number and size of the pores inside the samples change with the amount of F127 surfactant. The SLTO 0F127 sample (Fig. 3a and 3d) has no porosity,

and the grains have become agglomerated. In Fig. 3b and 3e, the SLTO 200F127 sample has few nanoscale porosities in between of grains. In the SLTO 600F127 sample, there are more mesopores in between of grains compare to the sample SLTO 200F127, as shown in Fig. 3c and 3f. It can be also observed from the fig 3 that the particle size increases with amount of surfactant in the sample. Increase in particle size could be the reason to keep the electrical conductivity unchanged but on the other hand the porosities in between of particles are responsible for phonon scattering which helps to reduce the phonon thermal conductivity.

The dependence on temperature of σ , S , PF , and κ for the La-doped SrTiO₃ samples with different amounts of F127 is shown in Fig. 4. The undoped SrTiO₃ is found insulator, however, its electrical conductivity has been improved with 20 at% La doping in each sample and this result is comparable with the previously reported results^{23, 24, 26, 33}. The electrical conductivity, σ of all the samples increases initially with temperature up to 647 K (Fig. 4a), and then it starts to decrease with temperature afterward. There is no substantial change in σ for the samples (Fig. 4a) with the different amount of F127 surfactant. The carrier concentration in all the samples is same since doping level is same for all the samples.

The Seebeck coefficient, S of all the samples is negative, and it increases in magnitude with temperature (Fig. 4b). There is a significant improvement in the Seebeck coefficient with increasing amounts of F127 surfactant which could be due to the scattering of charge carriers by the pore. The SLTO 600F127 sample shows a high Seebeck coefficient compare to other samples over a wide scale of temperature. The maximum value of the Seebeck coefficient for this sample is $140 \mu\text{V}\cdot\text{K}^{-1}$ at 325 K which is 52% higher than the sample without porosity (SLTO 0F127).

The power factor, PF for the samples is presented in Fig. 4c. Due to the improvement in the Seebeck coefficient, there is a significant improvement in the power factor also. The PF of

the samples increases with the temperature up to 647 K, where it has its peak value. The SLTO 600F127 sample shows the highest value of the power factor, $1.14 \text{ mW}\cdot\text{m}^{-1}\text{K}^{-2}$ at 647 K, which is 35% higher than the power factor of the sample without porosity (SLTO 0F127).

Fig. 4d exhibits the change in the thermal conductivity, κ of the samples with temperature. The thermal conductivity of the SLTO 0F127, SLTO 200F127 and SLTO 600F127 is found 3.03, 2.88, and $2.75 \text{ W}\cdot\text{m}^{-1}\cdot\text{K}^{-1}$ respectively at temperature 967 K. The gradual reduction in thermal conductivity has been observed with increasing amount of surfactant. Moreover, the κ of all the samples is found significantly lower than most of the published results^{25, 26, 34, 35}. The electronic thermal conductivity and the phonon thermal conductivity are presented in Fig S3 (a) and (b) respectively. The reduction in total thermal conductivity over the wide scale of temperature is due to the scattering of phonons by the nanoscale porosity. The Fig S3 b is the evidence of reduction in κ_{Ph} due to the nanoscale porosity.

Because of the substantial improvement in the Seebeck coefficient and reduction in the thermal conductivity, there is an overall improvement in the zT of the porous samples (SLTO 200F127 and SLTO 600F127) compare to the non-porous sample SLTO 0F127 as shown in Fig. 5a. There is also a substantial improvement in the zT over a wide scale of temperature compare to the previously reported result of 20 at% La doped SrTiO_3 ²³. The SLTO 600F127 sample shows the highest value of zT of 0.32 at 968 K.

The improvement in zT of the SLTO 600F127 in percentage compare to the sample SLTO 0F127 is shown in Fig. 5b. It is important to mention that, the average improvement of zT in the SLTO 600F127 is found 62% compare to the SLTO 0F127. The efficiency of samples is calculated according to the literature³⁶. The efficiency of samples SLTO 0F127 and SLTO 600F127 compare to the reported results is shown in Fig. 5c²³. It is found that the efficiency for the SLTO 600F127 is $>5\%$ at 968 K, which is around 26% higher than the SLTO 0F127 sample. The compatibility factor is important to cascade a thermoelectric material with another

one to fabricate segmented thermoelectric device. Two thermoelectric materials with close compatibility factor are suitable for cascading. The compatibility factor of samples (SLTO 0F127, SLTO 200F127, SLTO 600F127) for segmented thermoelectric generator is calculated based on the literature³⁷ as shown in Fig. 5d. It could help to find the suitable thermoelectric material for cascading with La-doped SrTiO₃.

Conclusions

La-doped SrTiO₃ bulk samples with F127 surfactant in different amounts have been fabricated for the first time and investigated successfully. The experiments reveal that there is an impact of nanoscale porosity on the transport properties of La-doped SrTiO₃. It has been observed that the Seebeck coefficient increases, while the thermal conductivity is reduced substantially by introducing porosity into the bulk sample because of the carrier and phonon scattering by the nanoscale pores. Therefore, there is an overall enhancement in the PF and the zT . The sample, SLTO 600F127, exhibits the highest value of the power factor, $1.14 \text{ mW}\cdot\text{m}^{-1}\text{K}^{-2}$ at 647 K, which is 35% higher than for the sample without porosity (SLTO 0F127). The same sample (SLTO 600F127) also exhibits the maximum value of the zT is 0.32 at 968 K with an average enhancement of 62% in zT in comparison to the sample without porosity (SLTO 0F127).

Experimental

Synthesis of La-doped SrTiO₃ (SLTO) powders with nanoscale porosity

First, strontium acetate (0.26 g) and lanthanum acetate hydrate (0.11 g for 20 at% La doping) were dissolved into acetic acid solution (3.0 mL) at 323 K with stirring. After the solution was cooled down to room temperature, titanium butoxide (0.61 g) was further added to it. The commercially available poly (ethylene oxide)-*b*-poly (propylene oxide)-*b*-poly (ethylene oxide) type triblock copolymer, Pluronic F127 (PEO₁₀₆PPO₇₀PEO₁₀₆), was used as the soft-template. Pluronic F127 in different amounts (200, and 600 mg) was dissolved in ethanol solution (3.0 g) in another beaker. The above two solutions were mixed together to prepare the precursor

solutions. Then, the precursor solutions were transferred onto filter paper. Finally, the filter papers were calcined for 10 min at 873K to remove the templates (Fig. 1).

Preparation of La-doped SrTiO₃ bulk

In the second stage of fabrication (Fig. 1), highly dense bulk samples of 20 mm diameter and around 1.5 mm thick (Table S1) were prepared from the La-doped SrTiO₃ calcinated powders by SPS (Thermal Technology SPS model 10⁻⁴) for 15 min at 1423 K with 60 MPa pressure in vacuum. Using a cutting machine (Struers Accutom-50), the bulk samples were shaped into rectangular bars and round disks for transport measurement. La-doped SrTiO₃ samples with different amounts of F127 surfactant were denoted by the amount of surfactant, such as SLTO-*x*F127 (*x* = 0, 200, and 600).

Sample Characterization

The room temperature powder XRD patterns were determined by the X-ray diffractometry (Cu K α , GBC MMA, $\lambda = 1.5418 \text{ \AA}$). XRD patterns were measured with a step size of 0.02° and speed of 2° per min from 10° to 80°. The morphologies and nanostructures of the powder and bulk samples were studied using field emission scanning electron microscopy (FE-SEM, JEOL 7500). The surface area and porosity of powder samples were inspected by Brunauer-Emmett-Teller (BET) and Barrett-Joyner-Halenda (BJH) analysis of nitrogen absorption-desorption data collected on a Tristar 3020 system (Micrometrics Instrument Corporation) after degassing at 150°C overnight. The *S* and the σ were measured from room temperature to 968 K under vacuum using Ozawa RZ2001i. The thermal diffusivity was measured under vacuum conditions using the instrument, LINSEIS LFA 1000, and the specific heat was measured under argon atmosphere by DSC-204F1 Phoenix. The weight and dimensions of a rectangular sample were used to determine the sample density. The results of the samples are confirmed by repeating all the measurements for several times.

Data Availability:

Data available at the Dryad Digital Repository: <https://doi.org/10.5061/dryad.4s6mn56> ³⁸

Competing Interests:

I/we have no competing interests.

Authors' Contributions:

Ahmed, Nazrul Islam, Ridwone Hossain and Billah carried out lab work, participated in data analysis and participated in the design of the study and drafted the manuscript. Jeonghun Kim, Minjun Kim and Shahriar Hossain carried out critically revised the manuscript. Yamauchi and Wang conceived of the study, designed the study, coordinated the study and helped draft the manuscript. All authors gave final approval for publication and agree to be held accountable for the work performed therein.

Funding:

This work was partially supported by the Australian Research Council (ARC) through a Discovery Project DP 130102956 (XLW), an ARC Professorial Future Fellowship project (FT 130100778, XLW), and a Linkage Infrastructure Equipment and Facilities (LIEF) Grant (LE 120100069, XLW). This research is also supported by the Global Connections Fund (Priming grant scheme) of the Australian Academy of Technology and Engineering (ATSE) in 2018, The University of Queensland's New staff start-up grant and funding received from Wollongong City Council

Research ethics:

Not required

Animal ethics:

Not required

Permission to carry out fieldwork:

No permissions were required.

Acknowledgements:

Higher degree research of Al Jumlat Ahmed and Sheik Md Kazi Nazrul Islam are supported by the Endeavour Leadership Program of Australian Government. We are thankful to Yaser Rehman, Hamzeh Qutaish and Alexander Morlando for assistance with sample characterization for SEM analysis and BET analysis.

References

1. H. Ohta, *Materials Today*, 2007, **10**, 44-49.
2. Y. Zhang, B. Feng, H. Hayashi, C.-P. Chang, Y.-M. Sheu, I. Tanaka, Y. Ikuhara and H. Ohta, *Nature Communications*, 2018, **9**, 2224.
3. X. Zhang and L.-D. Zhao, *Journal of Materiomics*, 2015, **1**, 92-105.
4. S. M. K. N. Islam, M. Li, U. Aydemir, X. Shi, L. Chen, G. J. Snyder and X. Wang, *Journal of Materials Chemistry A*, 2018, **6**, 18409-18416.
5. J. He and T. M. Tritt, *Science*, 2017, **357**.
6. J. R. Sootsman, D. Y. Chung and M. G. Kanatzidis, *Angewandte Chemie International Edition*, 2009, **48**, 8616-8639.
7. Y. Yin, B. Tudu and A. Tiwari, *Vacuum*, 2017, **146**, 356-374.
8. G. J. Snyder and E. S. Toberer, *Nature Materials*, 2008, **7**, 105.
9. L. Zhao, S. M. K. N. Islam, J. Wang, D. L. Cortie, X. Wang, Z. Cheng, J. Wang, N. Ye, S. Dou, X. Shi, L. Chen, G. J. Snyder and X. Wang, *Nano Energy*, 2017, **41**, 164-171.
10. Y. F. X. J. E. G. B. K. & and C. Z. I. F. N. Lu, *Adv Compos Hybrid Mater*, 2018, **2018**, 114-126.
11. Z. Liang, L. Ning, W. Hui-Qiong, Z. Yufeng, R. Fei, L. Xia-Xia, L. Ya-Ping, W. Xiao-Dan, H. Zheng, D. Yang, Y. Hao and Z. Jin-Cheng, *Chinese Physics B*, 2017, **26**, 016602.
12. H.-S. Kim, Z. M. Gibbs, Y. Tang, H. Wang and G. J. Snyder, *APL Materials*, 2015, **3**, 041506.
13. J. Tang, H.-T. Wang, D. H. Lee, M. Fardy, Z. Huo, T. P. Russell and P. Yang, *Nano Letters*, 2010, **10**, 4279-4283.
14. H. Lee, D. Vashaee, D. Z. Wang, M. S. Dresselhaus, Z. F. Ren and G. Chen, *Journal of Applied Physics*, 2010, **107**, 094308.
15. C.-S. Park, W. Han, D. I. Shim, H. H. Cho and H.-H. Park, *Journal of The Electrochemical Society*, 2016, **163**, E155-E158.
16. C.-S. Park, M.-H. Hong, H. H. Cho and H.-H. Park, *Journal of the European Ceramic Society*, 2018, **38**, 125-130.
17. M. Ohtaki, *Journal of the Ceramic Society of Japan*, 2011, **119**, 770-775.
18. H. Wang, W. Su, J. Liu and C. Wang, *Journal of Materiomics*, 2016, **2**, 225-236.
19. M. Li, D. L. Cortie, J. Liu, D. Yu, S. M. K. N. Islam, L. Zhao, D. R. Mitchell, R. A. Mole, M. B. Cortie and S. Dou, *Nano energy*, 2018, **53**, 993-1002.

20. M. Li, S. M. K. N. Islam, S. Dou and X. Wang, *Journal of Alloys and Compounds*, 2018, **769**, 59-64.
21. B. Shabbir, H. Huang, C. Yao, Y. Ma, S. Dou, T. H. Johansen, H. Hosono and X. Wang, *Physical Review Materials*, 2017, **1**, 044805.
22. B. Shabbir, X. Wang, Y. Ma, S. Dou, S.-S. Yan and L.-M. Mei, *Scientific reports*, 2016, **6**, 23044.
23. Z. Lu, H. Zhang, W. Lei, D. C. Sinclair and I. M. Reaney, *Chemistry of Materials*, 2016, **28**, 925-935.
24. R. Boston, W. L. Schmidt, G. D. Lewin, A. C. Iyasara, Z. Lu, H. Zhang, D. C. Sinclair and I. M. Reaney, *Chemistry of Materials*, 2017, **29**, 265-280.
25. A. Kikuchi, N. Okinaka and T. Akiyama, *Scripta Materialia*, 2010, **63**, 407-410.
26. K. Park, J. S. Son, S. I. Woo, K. Shin, M.-W. Oh, S.-D. Park and T. Hyeon, *Journal of Materials Chemistry A*, 2014, **2**, 4217-4224.
27. M. Qin, F. Gao, G. Dong, J. Xu, M. Fu, Y. Wang, M. Reece and H. Yan, *Journal of Alloys and Compounds*, 2018, **762**, 80-89.
28. Y. Wang and H. J. Fan, *Scripta Materialia*, 2011, **65**, 190-193.
29. H. Shen, Y. Lu, Y. Wang, Z. Pan, G. Cao, X. Yan and G. Fang, *Journal of Advanced Ceramics*, 2016, **5**, 298-307.
30. N. Suzuki, X. Jiang, R. R. Salunkhe, M. Osada and Y. Yamauchi *Chemistry – A European Journal*, 2014, **20**, 11283-11286.
31. N.-H. Park, F. Dang, C. Wan, W.-S. Seo and K. Koumoto, *Journal of Asian Ceramic Societies*, 2013, **1**, 35-40.
32. N. Stojilovic, *Journal of Chemical Education*, 2018, **95**, 598-600.
33. A. Kikuchi, L. Zhang, N. Okinaka, T. Tosho and T. Akiyama, *MATERIALS TRANSACTIONS*, 2010, **51**, 1919-1922.
34. D. Liu, Y. Zhang, H. Kang, J. Li, Z. Chen and T. Wang, *Journal of the European Ceramic Society*, 2018, **38**, 807-811.
35. J. Liu, C. L. Wang, Y. Li, W. B. Su, Y. H. Zhu, J. C. Li and L. M. Mei, *Journal of Applied Physics*, 2013, **114**, 223714.
36. H. S. Kim, W. Liu, G. Chen, C.-W. Chu and Z. Ren, *Proceedings of the National Academy of Sciences*, 2015, **112**, 8205-8210.
37. G. J. Snyder, *Applied Physics Letters*, 2004, **84**, 2436-2438.

38. Ahmed AJ, Nazrul Islam SMK, Hossain R, Kim J, Kim M, Billah MM, Hossain MS, Yamauchi Y, Wang X. Data from: Enhancement of Thermoelectric Properties of La-doped SrTiO₃ Bulk by Introducing Nanoscale Porosity. Dryad Digital Repository. <https://doi.org/10.5061/dryad.4s6mn56>

Footnote

† Electronic supplementary information (ESI) available: table